# Phenotypic Characterization of Colorectal Liver Metastases: Capsule versus No Capsule and the Potential Role of Epithelial Mesenchymal Transition

**DOI:** 10.3390/cancers15041056

**Published:** 2023-02-07

**Authors:** Claudia Fleig, Katja Evert, Hans J. Schlitt, Stefan Fichtner-Feigl, Stefan M. Brunner

**Affiliations:** 1Department of Surgery, University Medical Center Regensburg, 93053 Regensburg, Germany; 2Institute of Pathology, University Regensburg, 93053 Regensburg, Germany; 3Department of Surgery, University Medical Center Freiburg, 79106 Freiburg im Breisgau, Germany

**Keywords:** fibrous capsule, EMT, capsule components, protective border

## Abstract

**Simple Summary:**

Some colorectal liver metastases are surrounded by a fibrous capsule that has a positive impact on patient survival. This study aimed to characterize pathways that lead to this capsule formation. In colorectal liver metastases with capsule formation that consists of different collagen fibers, epithelial mesenchymal transition processes are upregulated and tumor cell migration is impaired. In summary, this leads to a less invasive tumor behavior when a capsule is present and can potentially be used to target tumor growth and spreading in the future.

**Abstract:**

**Background:** Colorectal liver metastases (CRLM) can be encased in a fibrous capsule separating cancer from normal liver tissue, which correlates with increased patient survival. This study investigated the cellular and molecular components of capsule formation and the possible role of epithelial mesenchymal transition (EMT). **Methods:** From 222 patients with CRLM, 84 patients (37.8%) were categorized to have CRLM encased with a capsule. A total of 34 CRLM from 34 selected patients was analyzed in detail by EMT pathway-profiling and custom PCR arrays to identify differences in gene expression between CRLM with (n = 20) and without capsule (n = 14). In parallel, those 34 CRLM were used to analyze 16 gene products at the metastasis margin via immunohistochemistry. **Results:** Encapsulated CRLM showed an elevated expression of signal transduction pathways and effector molecules involved in EMT. E-cadherin and keratin-19 were more prevalent, and transcription as well as translation (immunohistochemistry) of pGSK-3-β, SOX10, tomoregulin-1, and caldesmon were increased. By contrast, the loss of E-cadherin and the prevalence of snail-1 were increased in CRLM without capsule. Collagen I and III and versican were identified as capsule components with extracellular matrix fibers running concentrically around the malignant tissue and parallel to the invasive front. Caldesmon was also demonstrated as a capsule constituent. **Conclusions:** The fibrous capsule around CRLM can be produced by cells with mesenchymal characteristics. It functions as a protective border by both the features of fiber architecture and the inhibition of invasive growth through EMT recruiting mesenchymal cells such as myofibroblasts by transformation of surrounding epithelial or even carcinoma cells. By contrast, EMT demonstrated in non-encapsulated CRLM may lead to a more mesenchymal, mobile, and tissue-destructive carcinoma cell phenotype and facilitate malignant spread.

## 1. Introduction

Liver metastases of colorectal carcinomas may or may not be encircled by a fibrous capsule. Their prevalence range from 20 to 70% [1,2,3,4]. Not only does their simple presence indicate increased patient survival, but also their thickness has been shown to correlate with length of survival [4]. A 5-year patient survival between 39 and 88% was observed in patients with encapsulated CRLM, as compared to 19 to 34% in those CRLM without capsule. Furthermore, recurrence rates were reduced and R0 resection was achieved more frequently in patients with encapsulated CRLM [1,4,5,6,7]. Encapsulated metastases also seem to be better differentiated [4]. Brunner et al., found that localization (colonic versus rectal), infiltration depth, lymph node state and grading of the primary tumor, time of diagnosis (synchronous versus metachronous), and number and size of liver metastases as well as neoadjuvant chemotherapy within 3 months before resection were not significantly related to patient survival [1]. Immunohistochemically they identified collagen as a major capsule component and α-SMA-producing activated myofibroblasts were seen in the capsule area [1]. In addition, the presence of certain immune cells appeared to indicate better patient outcome. Cytotoxic T cells, Th1 cells, and memory cells are of advantageous in several types of carcinoma [8]. In CRLM, Brunner et al., described prolonged patient survival with CD45RO^+^, CD4^+,^ and CD8^+^ T cells accumulating around the primary tumor. The presence of such peritumoral T cells often coincided with the presence of a fibrous capsule around the hepatic metastases (47.7–83.5%) [1]. 

Characterizing the specific tumor micromilieu of these metastases is the first step to identifying the mechanisms underlying capsule production. Therefore, we investigated capsule components and analyzed capsule configuration. Furthermore, we addressed the possible role of epithelial mesenchymal transition (EMT) in the recruitment of cells capable to synthesize ECM for a fibrous capsule. To this end, various parameters indicating EMT activity were compared between CRLM with and without capsule. 

## 2. Methods

This study was performed at the University Medical Center of Regensburg, Germany, and was approved by the local ethics committee (no. 12-101-0009; 2012).

### 2.1. Materials

Fixated and paraffin-embedded CRLM and normal hepatic tissue were obtained from the Institute of Pathology of the University Hospital Regensburg, previously used and with protocols described in the study of Brunner et al. [1]. All antibodies were tested for high specificity and appropriate dilutions (Table 1). Patients (N = 222) were categorized as either with (N = 84; 37.8%) or without capsule formation (N = 138; 62.2%) without the category of partly encapsulated CLM. In the histological analysis, we decided according to the predominant characteristics in the largest available specimen. These two patient groups were comparable with regards to demographics besides a higher rate of lymph node metastasis of the primary tumor in the group of patients without capsule around their CRLM (Table 2). For the detailed analysis of capsule components, we selected a patient group that had a complete capsule and long survival in comparison to a group of patients who had no capsule with short survival. The rationale behind this selection was to enhance the opportunity to identify factors that are clinically relevant with regards to patient survival. Again, these two patient groups were comparable with regards to demographic factors and represented the real situation in clinical practice (Table 3). These paraffin blocks of metastatic and normal liver tissue were derived from the surgical samples of 34 selected patients who underwent potentially curative hepatic resection for CRLM between 2004 and 2010. Twenty CRLM had complete capsules confirmed by histology, also referred to as desmoplastic histologic growth pattern [9], while 14 were completely without capsule, also described as replacement histologic growth pattern [9].

### 2.2. Methods

To define the role of EMT in capsule formation, an EMT pathway-profiling PCR array was used as a screening method on three CRLM, each with and without capsule, respectively. As a reference, we used RNA of normal liver tissue of two of these patients. Based on these screening results, we selected 12 genes to be amplified in custom PCR arrays with another 28 CRLM to identify differences in gene expression between CRLM with (n = 17) and without capsule (n = 11). Paraffin blocks were cooled for 20 min at −20 °C and cut into 10 µm slices. Four slices of each block were put in an RNase free reaction tube (2 mL). RNA of CRLM was isolated using the *Purification of Total RNA from FFPE Tissue Sections* protocol (Qiagen, Hilden, Germany) [10]. To quantify isolated RNA, RNA samples were diluted with 1:30 DEPC water. Then, 60 µL of each sample were measured in a glass cuvette in a spectrometer at λ 260 nm. Isolated, non-contaminated RNA was transcribed to cDNA using the *First Strand cDNA Synthesis* protocol (Qiagen) [11]. The cDNA was preamplified before using the EMT PCR arrays, following the *Preamplification of cDNA Target Templates* protocol [11]. PCR Components Mix was prepared, and PCR [12] was conducted as described in the *Real-Time PCR Using RT^2^ Profiler PCR Arrays* protocol [11,13], using RT^2^ Profiler PCR Arrays (96-well) Human Epithelial to Mesenchymal Transition by Qiagen. The β2-microglobulin and glyceraldehyde-3-phosphate-dehydrogenase served as housekeeping genes. 

Implicating the results of our EMT PCR arrays, we chose 12 genes with altered gene expression in CRLM with and without capsule, Snail 1, Snail 2, Snail 3, pGSK-3-β, Caldesmon 1, TGF-β 1, TGF-β 2, Twist, Wnt 5a, Wnt 5b, Tomoregulin 1, and Nodal, to be quantified by qPCR in a Roche Light Cycler^®^ (Indianapolis, IN, USA) 480 via custom PCR arrays (Qiagen). On a 96-well plate of a custom array, six samples could be tested at once; summing up, gene expression in 28 other metastases was tested. The samples’ preparation for custom PCR arrays equaled the preparation for EMT PCR arrays except that the PCR Components Mix consisted of 17 µL of a preamplification product (sample), 212.5 µL of RT^2^ SYBR Green Mastermix, and 195.5 µL of RNase-free water.

The crossing points of EMT key genes were identified by Roche Light Cycler^®^ 480 Software and transferred into Microsoft Excel. Data were then uploaded on the PCR data analysis online program of Sabioscience. The cutoff point for PCR cycles was put at 35. Using the Sabioscience program, housekeeping genes and their arithmetic averages were determined and other genes were normalized. The average Ct and standard deviation of the genes of interest of the (1) capsule group and (2) no capsule group were determined. The ∆∆Ct calculation model [14] was used to analyze gene expression. The ratio of gene expression of the tested and the control group, normalized on reference genes, was presented as an n-fold change [15]. If the change was >twofold, gene expression was interpreted as upregulated in relation to the control group; if it was <twofold, gene expression was interpreted as downregulated. The significance level of the results was set at 5%. The *p*-values were calculated with normalized gene expression data using a Student’s *t*-test. The differences in regulation were presented in charts via Graphpad Prism 6. To analyze EMT arrays’ gene expression of CRLM with and without capsule, we compared it to the gene expression of normal liver tissue. 

In parallel, the components of the extracellular matrix in CRC and important gene products in the EMT were analyzed at the metastatic margin of the same 34 samples via immunohistochemistry or immunofluorescence using standard protocols (Table 1, 16 targets). Stained tissue was analyzed qualitatively by one observer with a Carl Zeiss Microscope Observer Z.1 and Zeiss AxioVision Rel. 4.8 Software. Slides were scanned with a Zeiss Mirax Desk Scanner and Mirax Desk Scan Software.

## 3. Results

### 3.1. Identification of Capsule Components and Analysis of Their Arrangement

In addition to α-SMA, collagen and CD45RO^+^, CD4^+^, and CD8^+^ T lymphocytes were detected in the capsule area by Brunner et al. [1]; collagen 1 (Figure 1A), collagen 3 (Figure 1B), and versican (Figure 1C) were identified as capsule components using immunohistochemistry. These intermediate filaments were also found in tumor septa (Figure 1A–C). In line with these findings, qPCR showed an upregulation of versican and collagen 1 and 3 in CRLM. Collagen 1 expression was seen more frequently in CRLM without than with capsule, while collagen 3 was expressed in a similar proportion in both groups. Fibronectin was visible in tumor septa but not in the capsule, where its mRNA levels were not elevated either. Collagen 5 is also not a capsule component, but it accumulated in perivascular connective tissue. Caldesmon, a protein-stabilizing actin filament, which, thus, minimizes cell motility and the invasive capability of carcinoma cells [16,17], was also identified in the capsule by immunohistochemistry (Figure 1D). Capsule filaments were arranged in a conformation parallel to the invasive tumor front (Figure 1A–D).

### 3.2. EMT Activity in CRLM with and without Capsule 

Based upon the above observations and the assumption that increased ECM expression would require recruitment of cells with mesenchymal phenotype, it was checked whether activity of EMT was upregulated in CRLM with as compared to without capsule and to normal hepatic tissue. 

#### 3.2.1. Transcription Factors Involved in EMT

Expression of Snail 1, 2, and 3, Sox 10, Twist 1, and Zeb 2 was found to be upregulated in CRLM with capsule, while Zeb 1 and Serpine 1 (plasminogen activator inhibitor) were not altered in comparison with non-capsulated CRLM or normal liver tissue (Figure 2A). Immunohistochemically, the Snail 1 gene product was present in non-tumor cells near the tumor margin of CRLM with capsule, while its expression in tumor cells was mostly observed in CRLM without capsule (Figure 3A). Snail 3 was shown in the nucleus of carcinoma cells close to the tumor margin of CRLM with capsule, while, in CRLM without capsule, Snail 3-stained tumor nuclei were both less frequent and only faintly dyed (Figure 3B). This observation corresponded with a Snail 3 mRNA level elevated eightfold in CRLM with capsule. Similarly, Sox 10 was immunohistochemically identified in tumor cells of CRLM with capsule, both membrane bound and in cytoplasm, correlating to the results of qPCR (Figure 3C), while Twist protein was seen in the cytoplasm of hepatocytes at the very margin of capsulated CRLM (Figure 3D).

#### 3.2.2. Proteins of TGF-ß Superfamily 

The expression of the activators of EMT transcription factors was also checked. BMP 2 and 7, Nodal, and TGF-β 2 and 3 were expressed more often in CRLM with capsule. In contrast, TGF-β 1 expression was similar in CRLM with and without capsule and normal liver tissue (Figure 2B).

#### 3.2.3. Essential Gene Products Involved in wnt Pathway

On the mRNA level, we also determined the expression of substantial proteins of the wnt pathway: β-catenin (Ctnnb1), wnt 11, 5a, and 5b, Gsk-3-β, and frizzled class receptor 7 (Fzd 7), which were all upregulated in CRLM with capsule (Figure 2C). Immunohistochemically, the analysis was focused on phosphorylated GSK-3-β and β-catenin. The pGSK-3-β was found preferentially in the cytoplasm of hepatocytes located close to the margin of non-encapsulated CRLM, while it was found in the cytoplasm of some tumor cells of CRLM with capsule (Figure 4A). In accordance with the activated wnt pathway, β-catenin was found to be accumulated in the nuclei of CRC cells immunohistochemically. While β-catenin could be demonstrated both at the cell membrane and in the nuclei of CRC cells located particularly at the tumor margin, it could only be identified at the cell membrane in hepatocytes (Figure 4B).

#### 3.2.4. Cell Junction Proteins

As a consequence of a potential increase in the expression of EMT-associated transcription factors induced by β-catenin, one might expect cell junctions to be reduced [18]. In fact, the expression of E-cadherin was decreased at the tumor margin both immunohistochemically (Figure 5A) and by immunofluorescence (Figure 5B). Its reduction was more prominent in non-encapsulated CRLM (Figure 5A,B, right).

The expression of cell junction proteins appeared to be increased in encapsulated as compared to non-encapsulated CRLM. Desmocollin 2, a part of desmosomes, and moesin, important for crosslinking plasma membrane and the actin cytoskeleton, were found more often in encapsulated metastases (Figure 2D). In addition, there was an increased transcription of intermediary filaments, keratin-7, -14, and -19 (Figure 2D). Keratin-19, which is positive in colorectal carcinoma, was immunohistochemically detected intercellularly in the entire tumor area (Figure 5C). Some extraneoplastic cells were also found positive for keratin 19 (Figure 5C). Notably, keratin-19 expression appeared to be reduced at the margin of non-encapsulated metastases (Figure 5C, right), while it was strongly expressed in encapsulated CRLM (Figure 5C, left). MAP1b, associated with modeling processes in the cytoskeleton (7) and absent in colonic carcinoma primary tumors (6), was equally expressed in metastatic and normal liver tissue on PCR. On immunohistochemistry, it was shown to be located in the cytoplasm of single cells around encapsulated CRLM (Figure 5D).

#### 3.2.5. Gene Products of Relevance for Extracellular Matrix

The transcription of gene products relevant for collagen and matrix was upregulated in both groups of CRLM as compared to liver tissue. Osteopontin mRNA (Spp1) was increased, especially in CRLM without capsule (Figure 2E). MMP 2, 3, and 9 were also elevated. While MMP 2 was equally expressed in both groups of CRLM, MMP 3 and 9 were more often translated at the tumor margin of encapsulated CRLM. (Figure 2E) TIMP-1, inhibiting the proteolysis of metalloproteinases [19], and versican were equally expressed in CRLM with and without capsule (Figure 2E).

#### 3.2.6. Tumor Suppressor Gene

The transmembrane protein tomoregulin-1 was detected in liver cells around encapsulated CRLM (Figure 6).

#### 3.2.7. Custom Arrays: Quantitative Validation of EMT Arrays

By custom PCR arrays of 28 patient samples, we could quantitatively validate the EMT PCR array results. In CRLM with capsule, Snail 1, 2, and 3, wnt 5a and 5b, Tomoregulin 1, Twist, Nodal, pGSK3B, and TGF-β 2 were upregulated. Caldesmon 1 and TGF-β 1 were equally expressed in CRLM and liver tissue. (Figure 2F)

In comparison to non-capsulated tumors, CRLM with capsule showed statistically significant upregulation of Snail 1, 2, and 3, wnt 5a and 5b, Tomoregulin 1, Nodal, and Twist (*p* < 0.02). Similarly, pGSK-3-β, Caldesmon 1, and TGF-β 2 expression were significantly increased (*p* < 0.03), although the difference was not regarded as being of biological relevance (fold regulation < 2). Concerning the expression of TGF-β 1, no difference was observed between both groups of CRLM (*p* > 0.05). 

## 4. Discussion

### 4.1. Activity of EMT at the Tumor Margin: Formation of a Fibrous Capsule Versus Invasive Tumor Growth 

In our present study, in CRLM with capsule, gene expression for products involved in EMT was demonstrated to be upregulated. 

The study found an increased expression of EMT transcription factors on both levels, mRNA and protein. Immunohistochemically, indicators of EMT occurred in different cell types (Figure 3A–D). Snail 3 as well as snail 1 were regarded as EMT inductors, although the precise role of snail 3 in EMT is still unclear. The injection of a snail 3 encoding retroviral construct into breast cancer cells triggered a mesenchymal phenotype and reduced invasiveness as compared to cells expressing snail 1 [20], much in support of our present data in CRLM. The role of Sox 10 is controversial depending on the type of cell and tissue observed. In our present study, it was detected in the cell membrane of cancer cells in CRLM with capsule, while it was absent in CRLM without capsule (Figure 3C). Supporting these findings, Tong et al., characterized SOX 10 function as a tumor suppressor gene in a study of digestive cancer cell lines including CRC as well as in an in vivo mouse system: interacting with β-catenin, it seemed to inhibit the wnt pathway, and, thus, proliferation and metastasis [21]. 

TGF-β may induce EMT through different mechanisms, leading to the recruitment of Snail 1 [20]. In our study, the expression of TGF-β 1 was similar in both groups of CRLM, while TGF-β 2 and 3 were increased in CRLM with capsule. 

Similarly, BMP 7 and E-cadherin were seen to be upregulated in encapsulated CRLM. BMP 7 is known as the endogenous counterpart of TGF-β-induced EMT. By the activation of transcription factors, it causes the production of E-cadherin [22,23,24]. Published information on the role of BMP in the development of carcinoma is controversial depending on the member of the BMP family and the type of tissue considered [25]. Due to these inconsistent results, its precise role in encapsulated CRLM remains speculative. 

Gene products of the wnt pathway were also upregulated in CRLM with as compared to without capsule. This finding was matched by the observation of increased transcription and intranuclear accumulation of β-catenin using PCR (Figure 2C) and immunohistochemistry (Figure 4B), respectively. 

Although the accumulation of β-catenin in nuclei and the loss of E-cadherin, particularly at the metastasis margin, were detected in both groups of CRLM, they were more apparent in CRLM without capsule. Therefore, increased EMT activity in this context would result in increased invasiveness [19]. 

Keratin 19 appeared to be significantly upregulated in both groups of CRLM, confirming its positivity in colorectal carcinoma [26]. The synthesis of keratin 7 (liver, [26]) and 14 (lung, [27]) was increased in the process of organ fibrosis. Rarely positive in CRC [28] and only marginally expressed in CRLM without capsule and normal liver tissue in our present study, they were substantially upregulated in encapsulated CRLM (Figure 2D). Whether this may be caused by some type of immune response, as suggested by the observation of immune cells around CRLM [1], leading to the activation of hepatic stellate cells, remains speculative. 

The expression of metalloproteases 2, 7, and 9 was upregulated in primary CRC [19,29,30] and seemed to be associated with poor patient outcome [30,31]. In our present study on CRLM, the expression of MMP 2, 3, and 9 was also increased in both groups of CRLM as compared to normal liver tissue. However, levels of MMP 3 and 9 expression were significantly higher in CRLM with than without capsule (Figure 2E). Since patients with encapsulated CRLM have longer survival, the expression of MMPs in this particular scenario seems to indicate EMT for capsule formation rather than increased invasiveness. 

### 4.2. Encapsulated Versus Non-Encapsulated CRLM: A Case of Carcinoma Cell Grading and Tumor Growth Potential

The calmodulin-, actin-, and myosin-binding protein caldesmon was immunohistochemically detected in the capsule area in our present study (Figure 1D, left). In addition to regulating the contraction of smooth muscle cells, caldesmon stabilizes actin filaments in non-muscle cells and is involved in cell shape modulation [16]. It is, therefore, considered to be a negative regulator of cell motility, proliferation, and MMP release by influencing the formation of podosomes. In human CRC cells, a reduction of invasive podosomes was shown by the exogen addition of caldesmon, while a depletion of caldesmon facilitated invasion in surrounding tissue [32]. Thus, the production of caldesmon in the capsule of CRLM, as demonstrated here, might decrease the invasive potential of CRC cells. In accordance with this finding, the detection of pGSK-3-β in the cytoplasm of CRC cells of encapsulated CRLM (Figure 4A) might indicate the increased degradation and ubiquitination of β-catenin. Consequently, tumor cell EMT and the potential to invade tissue would be reduced, similar to the results of a study on EMT in gastrointestinal carcinoma [33]. In colorectal carcinoma, keratin-19 positivity is a sign of better differentiated colorectal CRC [26]. While it was significantly upregulated in both groups of CRLM on the transcription level, immunohistochemistry demonstrated its presence throughout the entire metastasis in CRLM with capsule, while it was absent at the tumor margin of non-encapsulated CRLM (Figure 5C). This suggests that encapsulated CRLM exhibit a better grade of differentiation. 

### 4.3. Identified Capsule Components Suggest Mesenchymal Cells as a Potential Source 

Gene expression analysis of potential capsule components showed an upregulation of collagen 1, 3, and 5 in CRLM in comparison to liver tissue. Collagen 1 and 3 were also identified as capsule components by immunohistochemistry (Figure 1A,B), both shown to be synthesized by mesenchymal cells [34]. Increased collagen production was observed in CRC in vivo [35,36], most probably by the recruitment of myofibroblasts in the presence of carcinoma cells. Coulsen-Thomas et al., were able to demonstrate elevated production of collagen 1, 3, 4, and 5 and versican by fibroblasts in vitro, when they were co-cultured with and had cell–cell contact to carcinoma cells. The resulting thick extracellular matrix consistently reduced the migration of carcinoma cells [35].

In our study, versican was also seen to be upregulated in CRLM as compared to normal liver tissue. In adenocarcinoma, versican appeared to be produced by tumor-surrounding stroma cells or even the malignant cells themselves [37]. In accordance with its anti-adhesive properties and, thus, the facilitation of cell migration [37], an elevated level of versican has often been associated with a more invasive tumor phenotype [38,39,40]. Although this may also be operative in the case of CRLM, denseness, composition, and configuration of ECM might also play a critical inhibitory role for the invasiveness of carcinoma cells. In our present study, using immunohistochemistry, versican was found to represent a constituent of the fibrous capsule and of tumor septa. In accordance with the prolonged survival of patients with encapsulated CRLM, it might rather represent a product of mesenchymal cells and play a “protective” anti-invasive role. The evidence of intermediate filaments as capsule components strongly suggested that cells with mesenchymal properties were involved in capsule formation.

### 4.4. Matrix Architecture May Determine Capsule Function 

Carcinoma cells can easily move along ECM fibrils [37]. In breast cancer studies, the migration of carcinoma cells along collagen fibers perpendicular to the tumor front was shown and correlated with a more invasive phenotype [41]. As demonstrated in 2- and 3-dimensional co-culture systems of carcinoma cells and myofibroblasts, ECM accumulated around malignant cells; however, it reduced their migratory and invasive potential [35]. Therefore, a capsule composed of concentrically running matrix fibers may restrict the diffusion of tumor cell-derived soluble chemo attractants and growth factors and, thus, minimize the tumor’s invasive capability [35]. In our present study, collagen and other ECM fibrils assembling CRLM capsules ran parallel to the invasive tumor front. Thus, the filament texture, as found here, would likely enable tumor cells to move along the front between the tumor and capsule, while perpendicular movement away from the tumor might be restrained. As survival of patients with encapsulated CRLM had previously been shown to be increased [1], the configuration of collagen fibrils parallel to the tumor front has a clearly anti-invasive potential. Architectural arrangement of ECM fibrils, thus, seems to be of critical importance, either protecting the tumor-surrounding tissue from or facilitating the invasion of cancer cells. 

## 5. Conclusions

In conclusion, despite its clearly descriptive character, the present study gives important insights into the molecular pattern and architectural structure of the fibrous capsule enclosing CRLM and the potential mechanisms leading to its formation. EMT, as one of the mechanisms involved in the process of invasion into surrounding tissue and metastatic spread of CRC, seems to play an ambivalent role: on one hand, the transformation of the epithelial carcinoma cell into a more mesenchymal, mobile, and tissue-destructive phenotype is a prerequisite of malignant expansion, but, on the other hand, EMT leading to the recruitment of mesenchymal cells such as myofibroblasts by the transformation of surrounding epithelial cells or even carcinoma cells themselves and the formation of a fibrous capsule to encase CRLM might counteract malignant growth and further spread and, finally, improve patient survival. The various signal transduction pathways involved might, in the future, be investigated for their suitability as potential therapeutic targets.

## Figures and Tables

**Figure 1 cancers-15-01056-f001:**
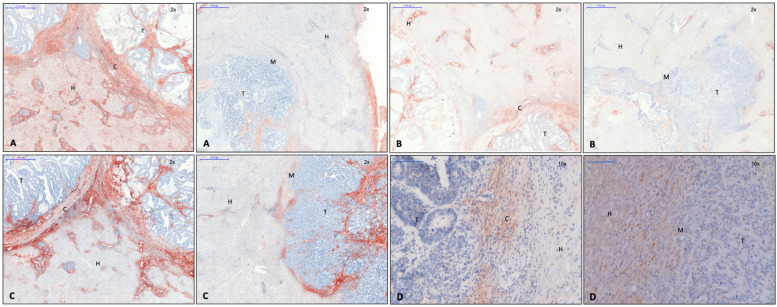
Left: capsule; right: no capsule. (**A**) Collagen 1 immunohistochemistry, 2× magnification. (**B**) Collagen 3 immunohistochemistry, 2× magnification. (**C**) Versican immunohistochemistry, 2× magnification. (**D**) Caldesmon immunohistochemistry, 10× magnification. Evidence of collagen 1 and 3, versican, and caldesmon in the fibrous capsule surrounding the metastasis. Legend: T, tumor; H, hepatocytes; M, tumor margin; C, capsule.

**Figure 2 cancers-15-01056-f002:**
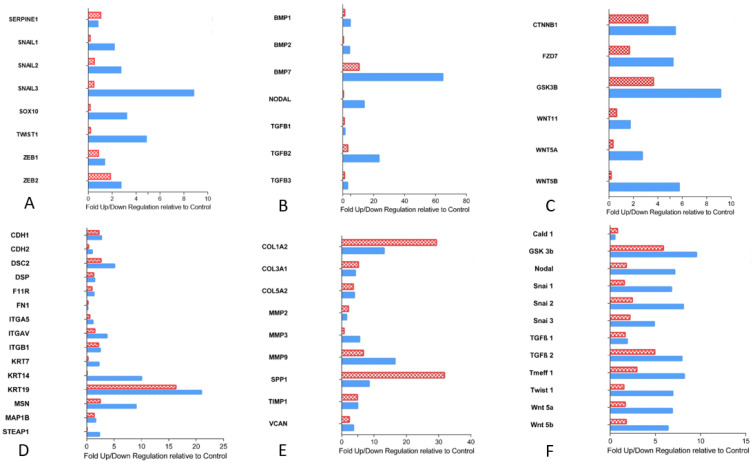
(**A**) Up- and downregulation of EMT transcription factors in CRLM with capsule (blue) and without capsule (red checked), diagrammed as multiple of liver tissue (control). (**B**) Up- and downregulation of TGF-ß superfamily in CRLM with capsule (blue) and without capsule (red checked), diagrammed as multiple of liver tissue (control). (**C**) Up- and downregulation of wnt pathway in CRLM with capsule (blue) and without capsule (red checked), diagrammed as multiple of liver tissue (control). (**D**) Up- and downregulation of relevant proteins for cell junctions in CRLM with capsule (blue) and without capsule (red checked), diagrammed as multiple of liver tissue (control). (**E**) Up- and downregulation of relevant gene products for collagen and matrix in CRLM with capsule (blue) and without capsule (red checked), diagrammed as multiple of liver tissue (control). (**F**) Up- and downregulation of 12 amplified genes in the custom arrays in CRLM with capsule (blue) and without capsule (red checked), diagrammed as multiple of liver tissue (control).

**Figure 3 cancers-15-01056-f003:**
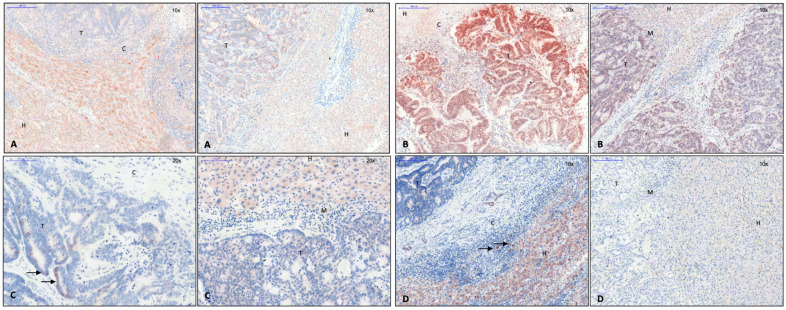
Left: capsule; right: no capsule. (**A**) Snail 1 immunohistochemistry, 10× magnification. Expression of Snail 1 in cytoplasm of cells near the tumor margin, mostly in encapsulated CRLM, and in cytoplasm of tumor cells, mostly in CRLM without capsule. (**B**) Snail 3 immunohistochemistry, 10× magnification. Expression of Snail 3 in nuclei of tumor cells at the margin of encapsulated CRLM. (**C**) Sox 10 immunohistochemistry, 20× magnification. Expression of SOX 10 at the membrane and in cytoplasm of tumor cells of encapsulated CRLM (arrows). (**D**) Twist immunohistochemistry, 10× magnification. Expression of Twist 1 in cytoplasm of cells close to the tumor margin in encapsulated CRLM (arrows). Legend: T, tumor; H, hepatocytes; M, tumor margin; C, capsule.

**Figure 4 cancers-15-01056-f004:**
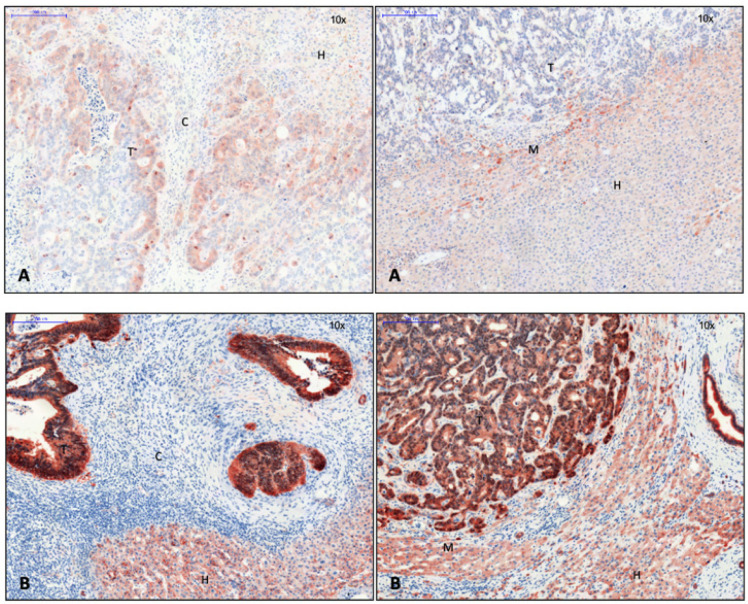
Left: capsule; right: no capsule. (**A**) The pGSK-3-ß immunohistochemistry, 10× magnification. Expression of pGSK-3-ß in cytoplasm of hepatocytes near the tumor margin in CRLM without capsule and in tumor cells at the tumor margin in CRLM with capsule. (**B**) The ß-catenin immunohistochemistry, 10× magnification. Expression of ß-catenin in nuclei of tumor cells, in membrane-bound hepatocytes. Cellular accumulation and deposit of ß-catenin in the nucleus of the tumor cell are associated with loss of E-cadherin and known as a phenomenon of EMT. Legend: T, tumor; H, hepatocytes; M, tumor margin; C, capsule.

**Figure 5 cancers-15-01056-f005:**
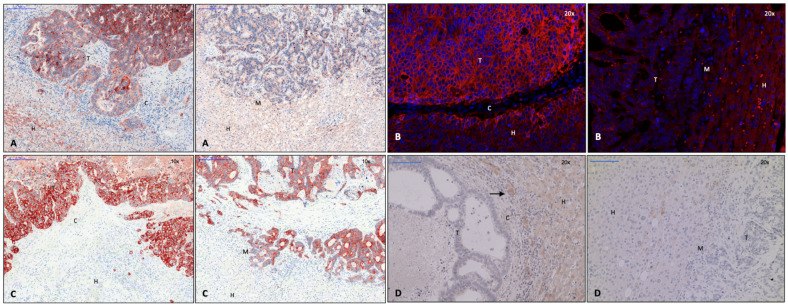
Left: capsule; right: no capsule. (**A**) E-cadherin immunohistochemistry, 10× magnification. Expression of E-cadherin is reduced at the tumor margin, especially in CRLM without capsule. (**B**) E-cadherin immunofluorescence, 20× magnification. Expression of E-cadherin is reduced at the tumor margin in CRLM without capsule. (**C**) Keratin-19 immunohistochemistry, 10× magnification. Intercellular expression of keratin-19 in the tumor area. In CRLM without capsule, expression is reduced at the tumor margin. (**D**) MAP1b immunohistochemistry, 20× magnification. Expression of MAP1b in cytoplasm of single liver cells surrounding the capsule (arrows). In CRLM, there was no evidence of MAP1b. Legend: T, tumor; H, hepatocytes; M, tumor margin; C, capsule.

**Figure 6 cancers-15-01056-f006:**
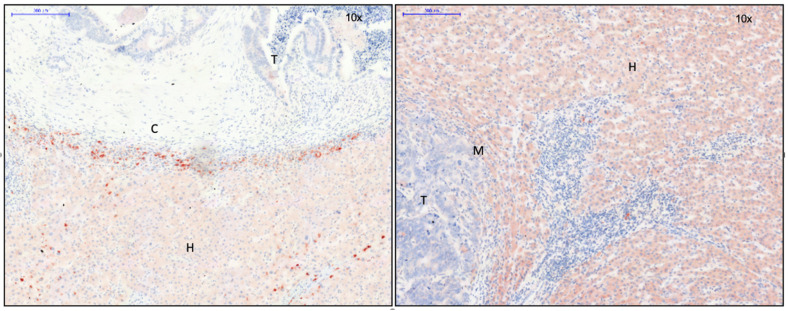
Left: capsule; right: no capsule. Tomoregulin-1, 10× magnification. Expression of tomoregulin-1 in liver cells near the tumor margin in CRLM with capsule. There was no evidence of tomoregulin-1 in CRLM without capsule. Legend: T, tumor; H, hepatocytes; M, tumor margin; C, capsule.

**Table 1 cancers-15-01056-t001:** Used antibodies.

Antibody	Host	Reactivity	Clonality	Dilution	Dwell Time AEC	Producer Catalogue Number	Concentration
β-Catenin	Rabbit	Human, Mouse, Rat, Ape, Zebrafish	Polyclonal	1:50	4 min	CellSignaling #9562	
Caldesmon 1	Rabbit	Human	Polyclonal, Clone: RB19749	1:10	1.5 min	Abgent AP6609c	0.25 mg/mL
Collagen 1	Mouse	Human, Rat, Cow, Game, Pig, Rabbit	Monoclonal, Clone: COL-1	1:200	2 min	Abcam Ab90395	
Col 3	Mouse	Human, Mouse, Rat	Monoclonal, Clone: Col-29	1:25	3 min	Abcam Ab82354	
Col 5	Mouse	Human, Sheep, Rabbit, Cow, Dog, Pig, Kangaroo	Monoclonal, Clone: 1E2-E4/Col5	1:25	3 min	Abcam Ab112551	
Cy3 Secondary antibody	Goat	IgG Rabbit	Polyclonal	1:100	-	Abcam Ab6939	
E-Cadherin	Rabbit	Human, Mouse	Monoclonal	1:100	3.5 min	CellSignaling 24E10, #3195	
Fibronectin 1	Mouse	Human	Monoclonal, Clone: IST-4	1:200	2.5 min	Sigma-Aldrich F0916	
Phospho-GSK-3 α, β (Ser21/9)	Rabbit	Human, Mouse, Rabbit, Ape, Zebrafish	Polyclonal	1:50	3 min	CellSignaling #9331	
Keratin 19 (BA17)	Mouse	Human	Monoclonal	1:200	2 min	CellSignaling #4558	
MAP1b	Mouse	Human, Rat, Cow	Monoclonal, Clone: 3G5	1:100	3 min	Abcam Ab79195	
Snail 1	Mouse	Human	Monoclonal	1:50	1 min	LifeSpanBioSciences, Inc. LS-C161335	
Snail 3	Rabbit	Human	Polyclonal	1:150	2 min	Sigma-Aldrich HPA016757	0.18 mg/mL
Sox 10	Mouse	Human	Monoklonal, Clone: 1E6	1:200	1 min	Sigma-Aldrich SAB1402361-100UG	1 mg/mL
Tomoregulin 1	Goat	Human, Mouse, Rat	Polyclonal	1:100	3 min	Biorbyt Orb101484	0.5 mg/mL
Twist 1	Mouse	Human	Monoclonal, Clone: 3E10	1:100	5 min	Abcam Ab135180	0.5 mg/mL
Versican	Mouse	Human	Monoclonal, Clone: 8.S.270	1:150	3 min	US Biological L1350A	~~1 mg/mL

**Table 2 cancers-15-01056-t002:** Demographics of investigated patients with colorectal liver metastases. * *p* < 0.05. Some histological results for T/N/G of primary tumors are missing.

	No Capsule (N = 138; 62.2%)	Capsule (N = 84; 37.8%)	*p*
Colon/Rectum	81/57	52/32	**0.673**
Synchronous/Metachronous liver metastasis	63/75	35/49	**0.580**
Singular/Multiple liver metastasis	51/87	33/51	**0.776**
No Chemo/Chemo (within 3 months prior to liver resection)	71/67	50/34	**0.268**
T1/2/3/4 (primary tumor)	6/11/89/29	1/14/55/13	**0.112**
N 0/1/2 (primary tumor)	35/51/49	40/23/20	**0.004 ***
G1/2/3 (primary tumor)	1/104/31	1/63/19	**0.939**

**Table 3 cancers-15-01056-t003:** Demographics of selected patients with colorectal liver metastases with and without capsule formation. * *p* < 0.05. Some histological results for T/N/G of primary tumors are missing.

	No Capsule (N = 14)	Capsule (N = 20)	*p*
Colon/Rectum	9/5	10/10	**0.495**
Synchronous/Metachronous liver metastasis	8/6	3/17	**0.023 ***
Singular/Multiple liver metastasis	4/10	8/12	**0.717**
No Chemo/Chemo (within 3 months prior to liver resection)	14/0	20/0	
T1/2/3/4 (primary tumor)	0/1/10/3	0/6/13/1	**0.133**
N 0/1/2 (primary tumor)	2/5/7	11/3/6	**0.051**
G1/2/3 (primary tumor)	0/11/3	1/15/4	**0.697**

## Data Availability

The data presented in this study are available on request from the corresponding author.

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
