# Peer review of "Phenotypic Characterization of Colorectal Liver Metastases: Capsule versus No Capsule and the Potential Role of Epithelial Mesenchymal Transition"

_cancers, 2023, doi:10.3390/cancers15041056_

Round 1
Reviewer 1 Report (Previous Reviewer 3)
The authors have now included additional tables with demographic information of the selected patients, which has made the study more complete. Although the study design is still relatively simple, the information this manuscript provides might be beneficial to some clinicians in this field.
Author Response
Our response:
We thank the reviewer for his expert revisions and the help to improve our manuscript.
Reviewer 2 Report (Previous Reviewer 2)
The authors have revised their manuscript.
Comments/Suggestions:
Table 2 and Table 3: Rektum -> Rectum.
Table 3: T 1/2/3/4, N 0/1/2, G 1/2/3: Please check the numbers presented in this table. The sums are not 138/84. (If data are not available for some of the patients, this should be stated.)
Author Response
Our response:
We thank the reviewer for the thorough revisions that further improved our manuscript. We changed rektum to rectum. Indeed, some histological results for the primary tumors (sometimes longer ago and not operated in our center) are missing. This is now correctly stated in the table lagends.
Reviewer 3 Report (New Reviewer)
Comments: The work is an interesting manuscript and in the scope of the journal field. However, there are some issues should be clarified
1. All immunohistochemistry results were provided with no negative slides. The purpose of the negative slides is to demonstrate whether the IHC test is specific. Usually, “negative slides” is a term applied to a separate slide with patient’s test sample on which no primary Antibody is applied.
2. The statistical analysis for gene expression was mostly not provided even standard error was missed. This causes a weak point to identify the significant value of gene expression.
3. Figure 2 is complicated and its quality so poor
4. In the section “protein TGF-β superfamily, TGF-β1,2 and 3 were investigated. However, the expression of TGF-βR1,2 was not confirmed.
5. There was no any evaluation of antibody markers in immunohistochemistry. The intensity of antibody marker and its presence should be evaluated to provide informatics analysis for each protein
Author Response
Our response:
Thank you for the thoughtful revisions of the manuscript. (1) In general, all antibodies used in our immunohistochemistries are established using a protocol that also employs negative slides, of course. In addition, multiple different concentrations are tested and the best options are selected. This is the normal process and we think, it is not necessary to add all these slides in the presentation of a manuscript. (2) In Figure 2 the statistical analysis is described in the methods section. (3) The two groups are calculated in a pooled fashion and normalized against normal liver tissue expression. This is the reason that there are no error bars and also no in depth statistical analysis. As described, we only depicted genes with at least a 2-fold in- or decrease. (4) Regarding the TGF superfamily, we indeed did not confirm TGF-βR1,2, but we think this would not add significantly to the overall message of the manuscript. (5) The reviewer is right, we did not quantify intensity of antibody marker and analyse this statistically using a computerized approach. From our point of view, it is better – especially for markers that are only present in very small areas or in case of only a few positive cells- to individually look at the pictures and describe the results. For example, it makes a huge difference, if positive cells are within the tumor or at the margin, if these cells migrate or not and where in relation to the fibrous capsule they are located. An analysis of overall intensity would fail to quantify and describe these really important features.
Round 2
Reviewer 3 Report (New Reviewer)
The manuscript is interesting and is more acceptable.
This manuscript is a resubmission of an earlier submission. The following is a list of the peer review reports and author responses from that submission.
Round 1
Reviewer 1 Report
This is a single-center retrospective study investigating the basic characteristics of the colorectal liver metastases (CLM) with or without capsule around tumors and the role of epithelial mesenchymal transition (EMT). The authors concluded that the fibrous capsule around CLM was produced by cells with mesenchymal characteristics and EMT could work for spreading malignant spread, however, the following concerns need to be addressed:
1. The number of samples analyzed in this study was small (n=34) and there was no statistical analysis.
2. It is well known that the capsule around CLM (i.e. desmoplastic layer) is not always observed in the whole rim of the tumor and it is impossible to dichotomize CLM according to the presence of capsule. The authors should state how they treated cases with CLM which have desmoplastic layer partly and show the result of intratumoral distribution regarding the gene expression and EMT.
Author Response
We thank the reviewer for his thorough review and hope to answer all pending questions:
To 1: The number of samples is small because we chose selected cases with capsule and long survival and, on the other hand, selected cases without capsule and short survival to better detect all potential differences between the groups. That is also to reason for not performing a statistical analysis because the groups were "artificial". This is now included in the manuscript.
To 2: As stated above in this analysis only patients with a complete caspsule and completely without capsule are included. When looking at all of our patients (not included in this study), of course, there are patients with partly encapsulated CRLM. In the other published study, these were counted to the respective group according to the predominant phenotype.
Reviewer 2 Report
Main Comments:
This manuscript deals with the potential role of epithelial mesenchymal transition in encapsulated and non-encapsulated colorectal liver metastases. The interactions are complex and, for the time being, many conclusions have to remain speculative, but – hopefully – it will be possible to translate the insights into further therapeutic concepts in the future.
There seems to be a discrepancy in the statements on Keratin-19: Figure legend 5C: "Intercellular expression of keratin 19 in the tumor area. In CRLM without capsule expression is reduced at the tumor margin." Lines 227-229: "Notably, keratin-19 expression appeared to be reduced at the margin of encapsulated metastases (Fig 5C, left), while it was strongly expressed in non-encapsulated CRLM (Fig. 5C, right)." Lines 324-329: "In colorectal carcinoma Keratin 19 positivity is a sign of better differentiated colorectal CRC (27). While it was significantly upregulated in both groups of CRLM on the transcription level, immunohistochemistry demonstrated its presence throughout the entire metastasis in CRLM with capsule, while it was absent at the tumor margin of non-encapsulated CRLM (Fig. 5C)." – Please clarify these passages.
Lines 306/307: "However, levels of MMP 3 and 9 expression were significantly higher in CRLM with than without capsule (figure 9)." – There is no "figure 9" in this manuscript.
Additional Comments:
List of abbreviations: All the abbreviations should be explained: SRY -> Sex-determining region Y (line 408), MMTV -> Mouse mammary tumor virus (line 414).
Figure legend 5 (line 212): "…properties It enables…" -> …properties. It enables…
Line 501 ("Figure Legends") is to be deleted.
Author Response
We thank the reviewer for his detailed review that detected some mistakes and misleading sentences.
We changed the manuscript according to the reviewers suggestions.
Reviewer 3 Report
This manuscript aimed to elucidate the possible mechanism involved in the capsule formation of colorectal liver metastases, and the authors used IHC/IFA as well as PCR-array analysis of FFPE samples from CRLM patients to study this.
1. A major limitation of IHC is that it depends on the high specificity of antibodies used and its signals are not easy to quantify. It’s not mentioned in the manuscript that if the authors have validated the antibodies for their specificity or if the antibodies have been verified previously. They should provide relevant results or discussion on this.
2. The demographic information such as the age & sex of the patients is missing.
3. The contents in Table1 are not well displayed and some items are not fully revealed. The authors might want to re-design the table so that readers can follow it more easily.
4. In line 207, the authors cited Figures 8A and 8B but I failed to find the two figures in the manuscript.
5. Based on the results in Figures 5A and 5B, it is difficult to draw the conclusion that “expression of E-Cadherin was decreased at the tumor margin”.
6. In line 210-212, the sentence “E-Cadherin is a component of cell junction…from the cell bond and migrate” need to be removed. It is not appropriate to include a detailed introduction in the figure legend.
7. Spelling errors and grammatical mistakes can be seen in the manuscript. For example, in line 73, “RNAse” should be “RNase”. In lines 78&82, “Quiagen” should be “Qiagen”.
Author Response
We thank the reviewer for his expert opinion and would like to answer to the raised questions.
To 1: This topic is of course important. We described the IHC methods in our previous paper, which we refer to in this paper. We have a standardized protocol to establish and ensure high specificity and optimal dilutions of all antibodies we use. This has been performed also in this case. This important information is now included in the mansucript.
To 2: The demographic data and also a statistical analysis of demographics is missing, because we intentionally selected cases with complete capsule and long survival versus cases completely without capsule and short survival to better detect potential differences. From our point of view, it does not make sense to analyse this highly selected patient group.
To 3: Table 1 is redesigned and included in the manuscript.
To 4, 6 and 7: We changed the mansucript according to the reviewers suggestions.
To 5: We think, that this pattern can be seen in the IHC, especially when looking at multiple images of all the patients. But since it is difficult to quantify and statistically analyse, we rewrote our statement in the mansucript in a less definitive way.
Round 2
Reviewer 1 Report
The authors answered to the reviewers’ questions, however, the following concerns still need to be addressed:
1. It is still hard to understand why the authors excluded patients with partly encapsulated CLM. As previously reported (Frantzas S. et al. Nat Med. 2016 Nov;22(11):1294), most of colorectal liver metastases (CLM) have partly capsule and patients included in this study should be a rare cohort. They should show the number of excluded patients. Additionally, if they wanted to show the eligibility of their conclusion, they should evaluate the distribution of cells with mesenchymal characteristics and gene expression related to epithelial mesenchymal transition in partly encapsulated CLM.
2. The authors stated their “artificial” stratification as the reason for the lack of statistical analysis, however it does not justify the lack of statistical analysis.
Reviewer 3 Report
The authors have made corresponding modifications to their manuscript based on my comments. However, there are still concerns about their response to comment 2. It is my understanding that the authors only selected the most typical cases to find more significant differences between the two phenotypes. And no statistical analysis could be processed under this circumstance. However, this leads to several major flaws in the experiment design: 1. Can this “highly selected group” represent the real situation in clinical practice? The experimental design does not seem to be scientifically sound. 2. One may naturally hypothesize that there will be differences between those two groups of patients, considering one is with capsule and the other isn’t. However, what’s more important and interesting is the mechanism underlying such difference. There is no mechanism study in this manuscript. The authors just verified that the difference between the two phenotypes did exist. The authors didn’t present enough data for their claim that “This study elucidates cellular and molecular mechanisms of capsule formation and the possible role of epithelial mesenchymal transition (EMT)” (line 12). The authors should include additional mechanism study results (in vivo or in vitro) to support their claim. Otherwise, such claims/statements should be revised or removed to accurately describe their current findings. For example, current study does not provide enough evidence for the statement that “mesenchymal cells as the MAIN source (of capsule formation?)” (line 334).
Minor comments: 1. The authors should add a scale bar to each image. Some of the 20x images seem to have the same magnification as the 10x ones. 2. The authors cited the same reference (#38) twice at the same place in line 347.